# Virulence beneath the fleece; a tale of foot-and-mouth disease virus pathogenesis in sheep

Carolina Stenfeldt[1,2], Juan M. Pacheco[1], Nagendrakumar B. Singanallur[3], Wilna Vosloo[3], Luis L. Rodriguez[1], Jonathan Arzt[1]*

1 Department of Agriculture, Foreign Animal Disease Research Unit, Agricultural Research Service, U.S., Plum Island Animal Disease Center, NY, Greenport, United States of America, 2 Department of Diagnostic Medicine/Pathobiology, Kansas State University, Manhattan, KS, United States of America, 3 Australian Animal Health Laboratory, CSIRO-Health and Biosecurity, Geelong, Australia

* Jonathan.Arzt@usda.gov

**Data Availability Statement:** All relevant data are within the paper.

**Funding:** This study was a collaboration between The Commonwealth Scientific and Industrial

## Abstract

Foot-and-mouth disease virus (FMDV) is capable of infecting all cloven-hoofed domestic livestock species, including cattle, pigs, goats, and sheep. However, in contrast to cattle and pigs, the pathogenesis of FMDV in small ruminants has been incompletely elucidated. The objective of the current investigation was to characterize tissue- and cellular tropism of early and late stages of FMDV infection in sheep following three different routes of simulated natural virus exposure. Extensive post-mortem harvest of tissue samples at pre-determined time points during early infection (24 and 48 hours post infection) demonstrated that tissues specifically susceptible to primary FMDV infection included the paraepiglottic- and palatine tonsils, as well as the nasopharyngeal mucosa. Additionally, experimental aerosol inoculation of sheep led to substantial virus replication in the lungs at 24–48 hours post-inoculation. During persistent infection (35 days post infection), the paraepiglottic- and palatine tonsils were the only tissues from which infectious FMDV was recovered. This is strikingly different from cattle, in which persistent FMDV infection has consistently been located to the nasopharyngeal mucosa. Analysis of tissue sections by immunomicroscopy revealed a strict epithelial tropism during both early and late phases of infection as FMDV was consistently localized to cytokeratin-expressing epithelial cells. This study expands upon previous knowledge of FMDV pathogenesis in sheep by providing detailed information on the temporo-anatomic distribution of FMDV in ovine tissues. Findings are discussed in relation to similar investigations previously performed in cattle and pigs, highlighting similarities and differences in FMDV pathogenesis across natural host species.

## Introduction

Foot-and-mouth disease (FMD) is an infectious disease of livestock that significantly impacts food security and agricultural economics in large regions of the world. FMD is endemic in most of Africa and Asia, while Europe, North America, Australia and New Zealand are kept

Research Organisation (CSIRO)-Australian Animal Health Laboratory (AAHL) and the Agricultural Research Service (ARS) of the US Department of Agriculture (USDA), who co-funded the work. Funding through CSIRO (NS, WV) was provided in part by the livestock industries in Australia through Animal Health Australia (AHA). The relevant industry bodies are the Cattle Council of Australia, Australian Dairy Farmers, Australian Lot Feeders Association, Wool Producers Australia, Sheepmeat Council of Australia, Australian Pork Limited and the Goat Industry Council of Australia. The AHA funds were matched through the Meat and Livestock Australia (MLA) Donor Company by the Australian Government under MLA Project P.PSH 0652. Additional funding came from USDA ARS (LR, JA) current research information system Project 1940- 32000-057-00D. The funders had no role in study design, data collection and analysis, decision to publish, or preparation of the manuscript.

**Competing interests:** The authors have declared that no competing interests exist.

free of FMD by employing strict regulation of import of animals and animal products. Potential incursions of the disease into countries that are normally free of FMD incur substantial costs from the drastic measures required for disease control, as well as prolonged losses of revenue from various aspects of animal production and trade [1].

FMD is caused by foot-and-mouth disease virus (FMDV), the prototype *Aphthovirus* within the *Picornaviridae* family. The virus is capable of infecting a wide range of cloven-hoofed domestic and wild species, although the severity of the disease may vary greatly depending on both host- and virus strain-specific factors [2, 3]. Clinical signs of FMD in sheep are often described as mild or inapparent [4]. However, some studies have reported severe clinical FMD in experimentally infected sheep characterized by fever, marked lameness, and vesiculo-erosive lesions on the feet and in the oral cavity [5, 6]. Additionally, FMD outbreaks may lead to substantial lamb mortality, which is generally attributed to FMDV-associated myocarditis [7, 8]. Still-birth and abortion have also been associated with FMD in sheep under natural [9] and experimental [10, 11] conditions.

From an epidemiological perspective, it has been shown that sheep may contribute substantially to dissemination of FMD outbreaks, specifically if infected sheep without apparent clinical signs of disease are relocated during early stages of an FMDV incursion to a previously free region [12]. Despite this, sheep are often excluded from preventative FMD vaccination efforts in endemic countries [13, 14], and the value of including sheep in emergency vaccination campaigns in the event of FMD outbreaks in previously free countries has been questioned [15].

Similar to other ruminant species, FMDV infection in sheep may lead to a persistent subclinical infection that has been reported to last for up to 9 months [16, 17], which is relatively short compared to cattle and buffalo [18–20]. Previous studies have suggested that in contrast to cattle, in which persistent FMDV infection has consistently been localized to the nasopharyngeal mucosa [21–24], persistent FMDV in sheep is more likely to be localized to the palatine tonsils [16, 25]. However, detailed information concerning the micro-anatomic localization and cellular tropism of persistent FMDV infection in sheep is lacking.

Experimental pathogenesis studies have demonstrated that the susceptibility to FMDV exposure via different routes, and the anatomic sites of primary infection are different in distinct host species; although this has only been thoroughly described in cattle and pigs. Specifically, while cattle are highly sensitive to virus exposure of the upper respiratory tract, with primary infection localized to the nasopharyngeal mucosa [26–28], pigs are more likely to become infected via oral exposure [29, 30], with primary infection occurring in epithelial crypts of tonsils within the oropharynx and laryngopharynx [31]. Additionally, it has been demonstrated that different systems for FMDV exposure of cattle may lead to slightly different temporo-anatomic progression during the very early stages of infection [26, 32]. Earlier investigations have shown that sheep are similar to cattle with regards to the sensitivity to FMDV exposure of the upper respiratory tract [5, 33–35]. A separate study involving FMDV detection in tissues harvested during early FMDV infection of adult ewes and lambs reported high viral loads in multiple tissues, including tonsils, lymph nodes, pharyngeal mucosa, lesion sites (tongue and coronary bands) and heart muscle from two days post-infection [8]. However, determination of the sites of primary infection was inconclusive due to the ubiquitously high viral loads in all sampled tissues, which is to be expected in tissues harvested from viremic animals.

The objective of this current investigation was to update the understanding of the anatomic distribution of FMDV in sheep during both early and late stages of infection, by use of experimental models optimized for FMDV pathogenesis studies in cattle and pigs. The current investigation included animals derived from two separate experimental studies originally designed for separate objectives: 1) evaluating different inoculation- and exposure systems for FMDV studies in sheep [5], and 2) investigating heterologous vaccine protection [36]. Sheep selected

for the current study were euthanized for post-mortem tissue harvest at pre-determined time points during early or persistent stages of infection. The output includes temporo-anatomic mapping of virus distribution at different times of infection as well as microscopic, cellular-level localization of FMDV within critical tissues.

## Materials and methods

### Virus

Foot-and-mouth disease virus (FMDV) O/SKR/2010 (O/SEA/Mya-98 lineage), kindly provided by Dr Kwang-Nyeong Lee, Ministry of Agriculture, Food and Rural Affairs, Republic of Korea, was originally derived from an FMDV-infected cow (NVRQS10, isolate 1012_49V) in Paju county, Gyeonggi province, the Republic of Korea in December of 2010 [37]. The field-derived virus was passaged once in cattle [38] before being used to infect sheep in this current study.

### Animals and animal experiments

The samples used for the current investigation originated from two separate experiments that have been previously published [5, 36]. Both experiments were carried out at the Plum Island Animal Disease Center (PIADC), New York. All experimental procedures were approved by the PIADC institutional animal care and usage committee (protocol 231-11-R) as well as the animal ethics committee of the Australian Animal Health Laboratory (AEC 1636). The sheep were approximately 6–12 months old crossbred Dorset males, delivered from a certified vendor.

**Experiment 1.** The objective of the first experiment was to investigate FMDV infection dynamics and virus distribution in tissues during the early stages of infection in sheep infected via simulated-natural inoculation. A detailed description of the study design and inoculation methods has been published previously [5]. In brief, groups of sheep were challenged with FMDV using one of four different systems; intra-nasopharyngeal (INP) inoculation, aerosol inoculation, coronary band injection, or contact exposure. To achieve contact exposure, four coronary band-inoculated sheep were moved into a different isolation room at 48 hours post-challenge (hpc) to function as virus donors to eight contact-exposed sheep. Two sheep from the INP- and aerosol-challenged groups were euthanized by intravenous injection of pentobarbital sodium at 24 and 48 hpc for detailed post-mortem tissue collection (Fig 1). Similarly, two sheep from the contact-exposed group were euthanized at each of 48 and 72 hpc for tissue collection. This current study focused only on the individuals that were euthanized for tissue harvest during early infection (n = 12). Due to practical limitations associated with the small group sizes, statistical analyses were not performed, and the output of these studies is purely descriptive and non-quantitative.

**Experiment 2.** Sheep for the second part of the study came from an earlier study that assessed the efficacy of a high-potency FMDV O1 Manisa vaccine (>6 50% Protective Doses [$PD_{50}$] per dose) against heterologous challenge [36]. These animals had received a single-dose of vaccine (1ml of a normal 2ml cattle dose) at 14 days before challenge (single dose/14d) or a double-dose of vaccine (2ml) at 7 or 14 days before challenge (double dose/7d, double dose/14d). All sheep were challenged by coronary band injection in one foot, and the clinical outcome of the study has been published earlier [36]. Detailed post-mortem tissue sampling was carried out on a subset of animals (n = 10) at 35–37 days post-challenge (dpc) for the objective of investigating tissue distribution of FMDV in sheep during persistent infection. The animal cohort used for post-mortem sampling included four non-vaccinated control animals, as well as two animals from each of the three vaccinated groups mentioned above. Vaccinated sheep

| | Intra-nasopharyngeal inoculation | | | | Aerosol inoculation | | | |
| | 24 HPI | | 48 HPI | | 24 HPI | | 48 HPI | |
| --- | --- | --- | --- | --- | --- | --- | --- | --- |
| *Animal ID* | *1301* | *1302* | *1304* | *1303* | *1309* | *1310* | *1312* | *1311* |
| *Clinical Score* | 0 | 0 | 0 | 2 | 0 | 0 | 0 | 1 |
| *Fever* | - | - | + | + | + | + | + | - |
| *FMDV RNA in serum* | 3.52 | 3.54 | 2.47 | 4.84 | 2.72 | 2.74 | 3.81 | 4.57 |
| **Oral cavity/ Oropharynx/ Laryngopharynx** | | | | | | | | |
| Dental pad | 3.15 | + | 3.08 | 4.50 | - | - | 6.59 | 8.56 |
| Tongue | 3.91 | + | 2.83 | 3.48 | - | - | 6.08 | 9.74 |
| Lingual tonsil | 3.63 | 3.51 | 2.16 | 4.43 | + | + | 6.17 | 5.57 |
| Palatine tonsil | 3.20 | 4.90 | 6.34 | 5.25 | - | + | 6.64 | 6.23 |
| Laryngal mucosa | 4.29 | 3.42 | + | 3.90 | 5.44 | 2.94 | 2.25 | 7.41 |
| Paraepiglottic tonsil | 5.69 | 3.99 | 2.42 | 3.51 | 3.92 | 3.13 | 5.23 | 6.07 |
| **Nasal cavity/ Nasopharynx** | | | | | | | | |
| Dorsal soft palate -Rostral | 4.58 | 4.35 | 4.06 | 3.97 | - | 3.05 | 5.96 | 5.64 |
| Dorsal soft palate -Caudal | 3.49 | 4.79 | 3.71 | 3.38 | 2.74 | 5.70 | 7.44 | 6.41 |
| Nasal turbinates | 3.39 | 2.96 | + | 2.44 | 4.06 | 3.35 | 7.24 | 6.20 |
| Nasopharyngeal tonsil | 3.25 | 5.77 | + | + | 2.51 | 3.35 | 7.52 | 6.24 |
| Dorsal nasopharynx -Rostral | 3.96 | 4.54 | 3.32 | 3.34 | 3.29 | 6.02 | 7.20 | 5.75 |
| Dorsal nasopharynx -Caudal | 3.77 | 4.23 | 4.09 | 3.82 | + | 4.89 | 3.86 | 6.39 |
| **Lungs** | | | | | | | | |
| Proximal cranial lobe | + | + | 2.14 | 4.41 | 5.29 | 6.17 | 6.22 | 7.68 |
| Middle cranial lobe | + | + | 2.82 | 3.89 | 6.18 | 6.90 | 6.00 | 8.33 |
| Distal cranial lobe | + | + | + | 3.50 | 7.21 | 6.18 | 6.40 | 8.55 |
| Proximal middle lobe | 2.78 | + | 3.30 | 4.06 | 7.04 | 6.18 | 5.83 | 7.38 |
| Middle middle lobe | + | + | 6.32 | 4.03 | 6.54 | 6.30 | 5.71 | 7.71 |
| Distal middle lobe | + | + | 4.83 | 4.14 | 6.52 | 6.76 | 4.36 | 8.39 |
| **Additional tissues** | | | | | | | | |
| Heart | - | *3.80* | - | *2.49* | *2.64* | + | *3.24* | 8.34 |
| Medial Retropharyngeal LN | 3.35 | 3.51 | 2.90 | 3.42 | + | + | 2.89 | 5.10 |
| Submandibular LN | 5.15 | + | 3.75 | 3.20 | - | + | 2.55 | 6.05 |
| Hilar LN | 2.73 | 2.64 | 2.44 | + | 4.46 | + | 2.72 | 7.82 |
| Renal LN | 2.81 | 2.80 | + | 3.07 | - | - | - | 5.98 |
| Popliteal LN | + | 2.90 | + | 3.05 | + | 6.01 | 4.12 | 5.04 |
| Interdigital cleft | 3.63 | 3.31 | 4.06 | 7.40 | - | - | 3.98 | 7.85 |
| Coronary band | 2.84 | + | 2.59 | 6.47 | 2.68 | - | 8.53 | 8.35 |

**Fig 1. FMDV distribution in ovine tissues during early infection.** FMDV detection in tissue samples obtained at 24 and 48 hours post intra-nasopharyngeal or aerosol inoculation. The clinical score represents the numbers of vesicular lesions detected at the time of euthanasia. Numbers in the table represent $\log_{10}$ genome copy numbers (GCN)/mg of FMDV RNA in tissue or $\log_{10}$ GCN/μl serum. Color gradient indicates increasing FMDV RNA quantities in samples that were positive by both RT-qPCR and virus isolation. Numbers in uncolored cells indicate quantities of FMDV RNA that were detected in samples that were negative by virus isolation, (+) indicates that virus isolation was positive but FMDV RNA content was below the limit of detection (<2.0 $\log_{10}$ GCN/mg), (-) indicates double-negative samples.

for post-mortem sampling were selected on the basis of consistent or intermittent detection of FMDV in oropharyngeal fluid (OPF) samples harvested between 14 and 35 dpc, with exception of the double dose/14d vaccine group in which there was no FMDV detection in OPF samples.

**Ante-mortem sample collection and clinical monitoring.** Whole blood samples were collected by jugular venipuncture at daily intervals during the early phase of infection (experiments 1 and 2), and weekly during later stages (experiment 2). Blood samples were centrifuged and serum was aliquoted and stored at -70˚C until further processing. Oropharyngeal fluid (OPF) samples were collected using a probang cup adapted for small ruminants [16], twice weekly from 14 to 35 dpc in experiment 2. The probang cup was rinsed in 2ml of minimal essential media containing 25 mM HEPES, which was further separated into aliquots. Prior to

virus isolation, one aliquot of OPF was treated with 1,1,2-trichlorotrifluoroethane (TTE; Sigma-Aldrich) for dissociation of potentially antibody-complexed virus [23, 39]. The progression of clinical FMD during early infection (1–10 dpc) was monitored using a cumulative lesion score as previously described [5]. In brief, vesicular lesions at medial or lateral aspects of the two main digits of each foot contributed one point each (maximum 4 points per foot), with additional single points counted for lesions on the dental pad, tongue, lips and nostrils, leading to a total maximum score of 20. For sheep infected by coronary band injection, the inoculated foot was excluded from scoring, giving a maximum score of 16.

**Post-mortem tissue collection.** A standardized necropsy procedure with collection of 18–22 distinct tissue samples was performed immediately after euthanasia. Tissues of specific interest included anatomic sites that have been previously indicated in FMDV pathogenesis in sheep and other susceptible host species. These included the nasopharyngeal mucosa which is comprised of the contiguous epithelial surfaces of the dorsal aspect of the soft palate and the adjoining dorsal nasopharynx, as well as the nasopharyngeal-, palatine-, and paraepiglottic-tonsils, which are distinct tonsils situated in the nasopharynx, oropharynx and laryngopharynx, respectively. Each tissue sample was divided into 30mg aliquots which were placed in individual tubes before being frozen over liquid nitrogen vapour. An adjacent specimen from each tissue was divided into two replicates that were embedded in optimal cutting temperature media (Sakura Finetek, Torrance, CA) in cryomolds and frozen over liquid nitrogen vapour. Tissue samples were kept frozen in the vapour phase over liquid nitrogen, and were transferred to the lab within two hours of collection for storage at -70° C until further processing.

## FMDV RNA detection

Two aliquots of each tissue sample collected at necropsy were thawed and individually macerated in tissue culture media, using a TissueLyser bead beater (Qiagen, Valencia, CA) and stainless steel beads (Qiagen cat. no. 69989). Total RNA was extracted from tissue macerates, serum, and OPF samples using Ambion's MagMax-96 Viral RNA Isolation Kit (Ambion, Austin, TX) on a King Fisher-96 Magnetic Particle Processor (Thermo Scientific, Waltham, MA). Extracted RNA was analyzed using quantitative real-time RT-PCR (RT-qPCR), targeting the 3D region of the FMDV genome [40] with forward and reverse primers adapted from Rasmussen et al [41], and chemistry and cycling conditions as previously described [42]. Cycle threshold values were converted to FMDV RNA copies using an equation derived from analysis of serial 10-fold dilutions of *in vitro* synthesized FMDV RNA of known concentration. The equations of the curve of RNA copy numbers versus Ct values were further adjusted for the average mass of tissue samples and specific dilutions used during processing of samples.

## Virus isolation

Aliquots of macerated tissue samples and TTE-treated probang samples were cleared from debris and potential bacterial contamination by centrifugation through Spin-X filter columns (pore size 0.45μm, Sigma-Aldrich) and were subsequently analyzed for infectious FMDV by virus isolation (VI) on LFBK-αvβ6 cells [43–45], following a protocol previously described [28]. All VI cell culture supernatants were analyzed by RT-qPCR, as described above, to confirm the presence or absence of amplified FMDV.

## Immunomicroscopy

Detection of FMDV antigen in cryosections by immunohistochemistry (IHC) and multichannel immunofluorescense (MIF) was performed as previously described [26, 46]. Slides were examined with a wide-field, epifluorescent microscope, and images were captured with a

cooled, monochromatic digital camera. Images of individual detection channels were adjusted for contrast and brightness and merged in commercially available software (Adobe Photoshop CC2019). Alternate sections of analyzed tissues were included as isotype controls, and additional negative control tissue sections were prepared from corresponding tissues derived from non-infected animals. Detection of FMDV antigen was performed using mouse monoclonal antibodies against structural proteins (VP1; F1412SA [47]) and nonstructural proteins (3D; F19-6(302)[48]). MIF experiments included labelling of cell markers using rabbit polyclonal anti-pancytokeratin (Invitrogen 180059), mouse monoclonal anti-sheep MHC II (Serotec MCA2228), and mouse monoclonal anti-CD11c (Washington State University, clone BAQ153A).

## Results

### Early infection

For investigation of FMDV distribution in ovine tissues during early stages of infection, 12 sheep that had been infected through intra-nasopharyngeal (INP) inoculation, aerosol inoculation, or contact exposure were euthanized for tissue harvest at 24–72 hpc. Detailed descriptions of the clinical progression of FMD in these experiments have been published previously [5]. In brief; there were no clinical signs and no detection of FMDV in any samples obtained from the 4 sheep from the contact-challenged group that had been randomly selected for post-mortem tissue sampling at 48 and 72 hpc. Amongst the 4 INP-inoculated sheep selected for post-mortem sampling, there were no clinical signs of FMD at 24 hpc. At 48 hpc, the two surviving individuals had rectal temperatures >40˚C, and one animal had two small vesicles on the dental pad and in the interdigital cleft of one hoof (Fig 1, animal ID 1303). In the aerosol-inoculated group, 3 of 4 individuals had pyrexia (>40˚C) at 24 hpc, but no vesicular lesions were observed. At 48 hpc, one of the two remaining animals (ID 1311) had a vesicular lesion on the tongue. All sheep from the INP- and aerosol inoculated groups were viremic at the time of euthanasia and tissue harvest.

### Anatomic distribution of FMDV during early infection

**Intra-nasopharyngeal inoculation.** The highest quantities of FMDV RNA at 24 hpc in INP-inoculated sheep were found in the tonsils of the upper respiratory and gastrointestinal tracts (Fig 1). The paraepiglottic-, nasopharyngeal-, and palatine tonsils contained FMDV RNA quantities that exceeded concurrent quantities in serum (4.90–5.77 $\log_{10}$ genome copy numbers (GCN)/mg). Slightly lower levels of viral RNA were detected in the mucosa of the dorsal surface of the soft palate, and the adjoining dorsal nasopharynx (Fig 1). Although infectious FMDV was isolated from all pulmonary samples harvested at this time point, FMDV RNA quantities in lung samples were below the limit of detection (<2.0 $\log_{10}$GCN/mg) of the RT-qPCR assay in all but one sample (Fig 1). At 48 hpc, virus distribution was more disseminated, with detection in almost all tissues sampled. The highest quantity of virus at this time point was in the vesicular lesions of the interdigital cleft (7.40 $\log_{10}$GCN/mg) and coronary band (6.47 $\log_{10}$GCN/mg) of sheep number 1303. Other samples with remarkably high RNA copy numbers were the palatine tonsil and the middle pulmonary lobe of sheep 1304 (6.32–6.34 $\log_{10}$GCN/mg, Fig 1).

**Aerosol inoculation.** In contrast to INP-inoculation, aerosol inoculation resulted in markedly higher quantities of FMDV RNA in the lower respiratory tract compared to the upper respiratory tract at 24 hpc. Viral RNA quantities measured in lung samples at this early time point (up to 7.21 $\log_{10}$ GCN/mg) far exceeded concurrent serum levels (2.72–2.74 $\log_{10}$GCN/μl; Fig 1). At 48 hpc, virus quantities in samples from the upper respiratory- and

gastrointestinal tracts were comparable to those detected in the lower respiratory tract. The highest FMDV RNA quantity was found in a tongue lesion of animal number 1311 (9.74 $\log_{10}$GCN/mg; Fig 1). Despite absence of macroscopic lesions, very high levels of FMDV RNA were also found in samples from other lesion predilection sites, such as the coronary bands and dental pad, suggesting that lesion development was possibly underway at these sites. Comparably high viral RNA quantities (8.33–8.55 $\log_{10}$ GCN/mg) were detected in the myocardium and lung samples of animal 1311 (Fig 1).

## Immunomicroscopic detection of FMDV in ovine tissues during early infection

Screening of tissue sections by immunohistochemistry localized early FMDV infection to the paraepiglottic-, nasopharyngeal-, and palatine tonsils following INP inoculation. At 24 hpc, FMDV antigen was present in intra-epithelial micro-vesicles within the surface epithelium of the paraepiglottic tonsil, further emphasizing the relevance of this tissue as a site of FMDV replication during early infection in sheep (Fig 2A and 2B). These microvesicles had typical features of FMD vesicles including central cavitation and epithelial acantholysis. At 48 hpc, virus replication was localized to epithelial crypts of the palatine tonsils, as demonstrated by detection of high quantities of FMDV non-structural antigen (Fig 3C and 3D).

In the aerosol-inoculated sheep, FMDV antigen was predominantly detected in pulmonary tissue samples during early infection (Fig 3C and 3D). Within the lungs, FMDV antigens were detected in discrete foci within the pulmonary parenchyma. Viral antigens were localized to cytokeratin-expressing epithelial cells which were either intact within the alveolar septa or acantholytic within alveolar lumina (Fig 3C and 3D). Additional evidence of primary virus replication was localized to the dorsal nasopharyngeal mucosa at 24 hpc (Fig 3A and 3B) as well as epithelial crypts of the palatine tonsil at 48 hpc (not shown). Within the nasopharynx, FMDV antigen detection was associated with a focal erosion within a region of lymphoid-associated epithelium of the mucosal surface (Fig 3A and 3B).

## Persistent infection

The investigation of FMDV persistence in sheep was based on a total of 10 animals. This subset of animals included four non-vaccinated controls, and two sheep from each of the three vaccinated cohorts: single-dose vaccine at 14 days before challenge, double dose vaccine at 14 days before challenge, and double dose vaccine at 7 days before challenge [36]. The 4 non-vaccinated sheep all developed moderate to severe clinical FMD, with maximum clinical scores ranging from 9–15 (out of a maximum score of 16; Fig 4). The two sheep that had received a single dose of vaccine at 14 days prior to challenge were protected from clinical FMD, whereas the four sheep that had received a double vaccine dose at 7 or 14 days prior to challenge all had mild clinical FMD after challenge (clinical score range 2–7; Fig 4).

**Detection of FMDV in OPF.** The vaccinated animals from experiment 2 that were included in this study were selected on the basis of detection of FMDV RNA and/or infectious FMDV in OPF samples harvested between 14 and 35 dpc. Additionally, two vaccinated sheep with consistently negative OPF samples (IDs 1367 and 1368) were included in this investigation as representatives of the study group that had received a double dose vaccine at 14 days prior to challenge, as there were no OPF-positive animals within that group. Three of the four sheep in the non-vaccinated group (IDs 1376, 1377, and 1378) had consistent detection of FMDV RNA, with concurrent isolation of virus from the majority of OPF samples harvested from 14 to 35 dpc (Fig 4). There were no FMDV-

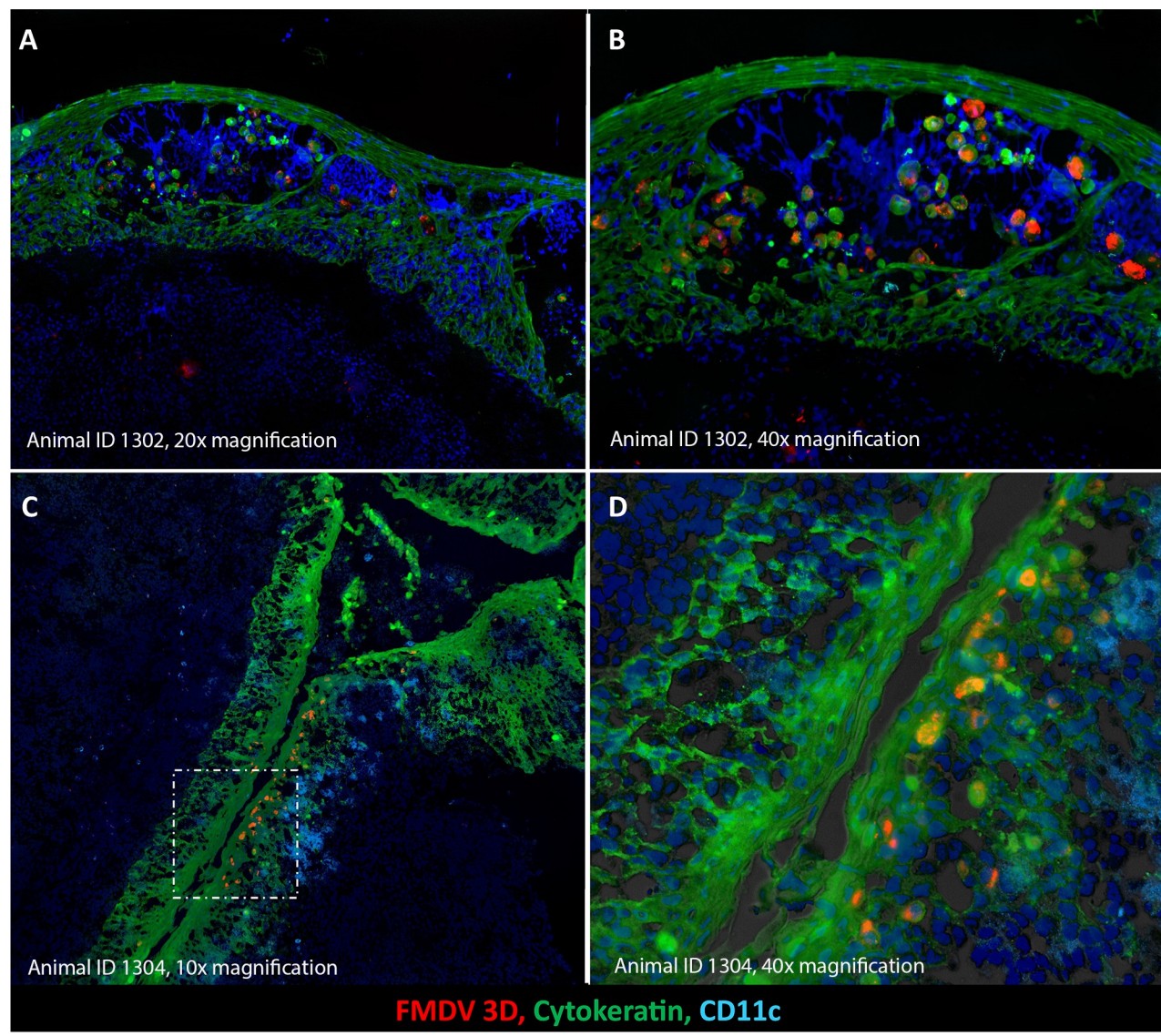

**FMDV 3D, Cytokeratin, CD11c**

**Fig 2. FMDV infection of paraepiglottic- and palatine tonsils of sheep during early stages of disease in INP-inoculated sheep.** A-B) Primary FMDV infection of the paraepiglottic tonsil at 24 hours post intra-nasopharyngeal inoculation (animal ID 1301). A microvesicle within the surface epithelium of the tonsil contains large quantities of acantholytic, FMDV 3D/cytokeratin double-positive epithelial cells. A = 20x magnification, B = 40x magnification. **C-D)** FMDV-replication in an epithelial crypt of the ovine palatine tonsil at 48 hours post intra-nasopharyngeal inoculation (animal ID 1304). Large quantities of FMDV 3D (red) are present within structurally intact, cytokeratin+ (green) epithelial cells within the tonsil crypt. C = 10x magnification. D = 40x magnification with differential interference contrast.

positive OPF samples from the fourth individual of the non-vaccinated group (ID 1379). In the single dose/14d vaccine group, OPF samples from one individual (ID 1361) were positive by RT-qPCR and VI at 17, 24, and 28 dpc. The second individual in that group (ID 1355) had five RT-qPCR-positive OPF samples from 14 to 28 dpc, with VI-positive samples at 17, 21, 28, and 31 dpc (Fig 4). Five of the seven OPF samples obtained from sheep 1373 in the double dose/7d vaccine group were RT-qPCR/VI double-positive, whereas the second individual of that group (ID 1369) had only one RT-qPCR-positive/VI-negative sample (Fig 4)

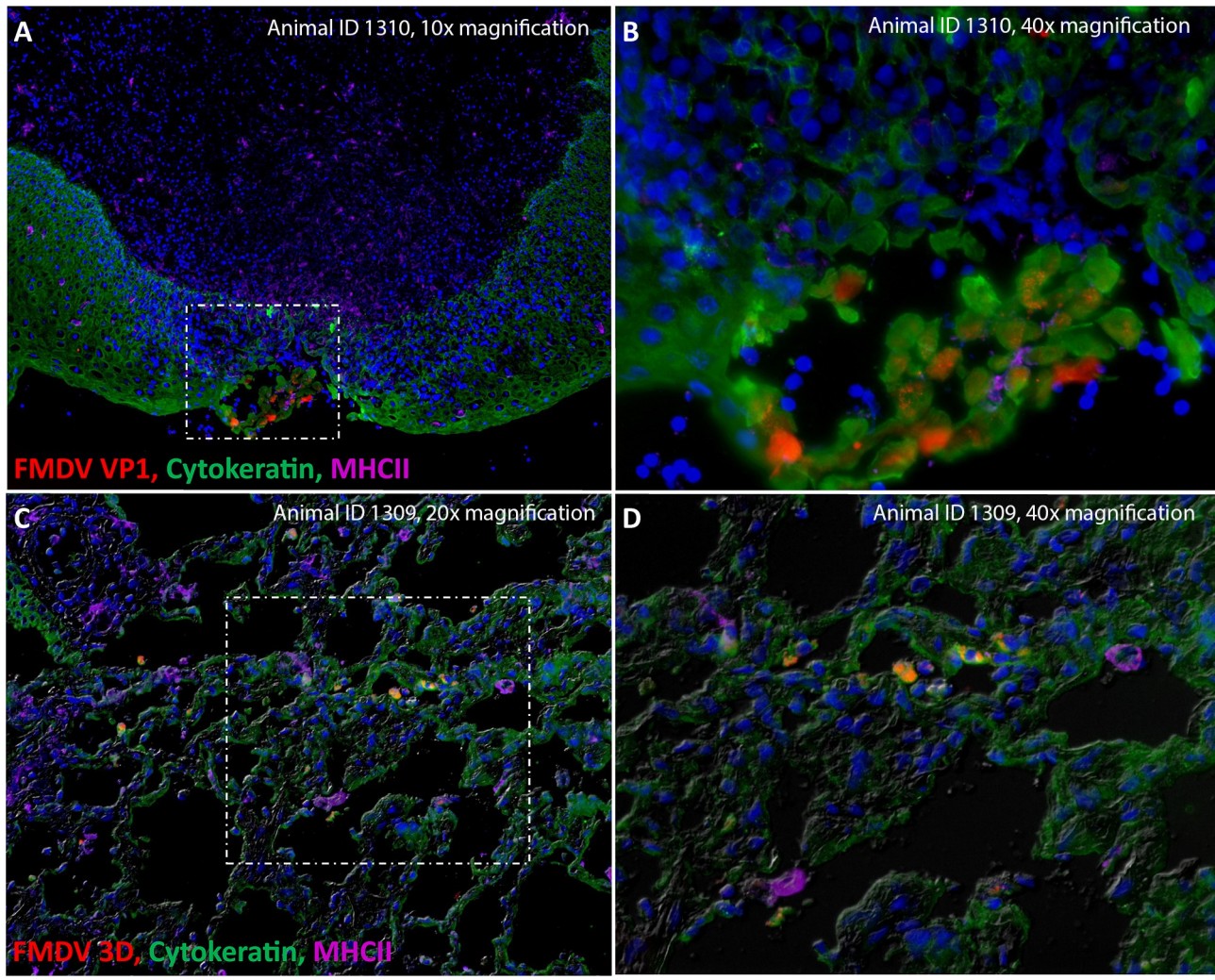

**Fig 3. FMDV infection in ovine nasopharyngeal mucosa and lungs during early stages of disease in aerosol-inoculated sheep.** A-B) FMDV infection in the dorsal nasopharyngeal mucosa at 24 hours post aerosol inoculation (animal ID 1310). FMDV VP1 (red) is localized to cytokeratin+ epithelial cells (green) within a surface erosion in a segment of MALT-associated epithelium. MHCII+ cells are abundant throughout the submucosa and are infiltrating the area of the lesion. A = 10x magnification. B = 40x magnification. **C-D)** Microscopic distribution of FMDV non-structural protein 3D (red) in lungs of FMDV-infected sheep at 24 hours post aerosol inoculation (animal ID 1309). FMDV 3D (red) predominantly localizes within intact and acantholytic cytokeratin-positive pulmonary epithelial cells (green), but not with MHC II (purple) expressing host cells. C = 20x magnification. D = 40x magnification with differential interference contrast.

## Detection of FMDV in ovine tissues during persistent infection

Infectious FMDV was isolated from samples of the palatine tonsils harvested at 35 dpc, with concurrent detection of FMDV RNA, from three of the four non-vaccinated sheep, and one of two sheep in the single dose/14d vaccine group (Fig 4). Additionally, the paraepiglottic tonsil from one of the non-vaccinated sheep was RT-qPCR/VI-positive. There was no detection of FMDV RNA or isolation of virus from any tissue harvested from the two sheep that had received a double vaccine dose 14 days before challenge. In the remaining vaccinated groups, there was consistent detection of FMDV RNA, but no infectious FMDV in multiple samples of palatine- and paraepiglottic tonsils and single samples of the dorsal soft palate and medial retropharyngeal lymph node. FMDV RNA detection was more disseminated within the non-

| | Non-Vaccinated | | | | Vaccinated | | | | | |
| --- | --- | --- | --- | --- | --- | --- | --- | --- | --- | --- |
| | | | | | *single dose 14d* | | *double dose 14d* | | *double dose 7d* | |
| *Animal ID* | *1376* | *1377* | *1378* | *1379* | *1355* | *1361* | *1367* | *1368* | *1369* | *1373* |
| *Highest clinical score (early infection)* | **10** | **9** | **14** | **15** | **0** | **0** | **7** | **2** | **5** | **2** |
| **Oral cavity/ oropharynx** | | | | | | | | | | |
| Lingual tonsil | - | - | - | - | - | - | - | - | - | - |
| Paraepiglottic tonsil | 3.68 | 2.36 | 3.46 | - | 3.49 | - | - | - | 2.79 | 2.36 |
| Palatine tonsil | 3.06 | 5.30 | 4.67 | - | 3.73 | 2.61 | - | - | 3.39 | 3.50 |
| **Oropharynx/ Nasopharynx** | | | | | | | | | | |
| Ventral soft palate | + | 2.52 | 3.96 | - | - | - | - | - | - | - |
| Dorsal soft palate | - | 3.39 | 3.09 | - | - | - | - | - | - | 2.84 |
| Nasopharyngeal tonsil | 2.46 | - | 2.05 | - | - | - | - | - | - | - |
| Dorsal nasopharynx | - | - | - | - | - | - | - | - | - | - |
| **Additional tissues** | | | | | | | | | | |
| Eosophagus | - | - | - | - | - | - | - | - | - | - |
| Lung | - | - | - | - | - | - | - | - | - | - |
| Medial Retropharyngeal LN | 3.41 | - | - | 2.04 | - | - | - | - | 2.39 | - |
| Submandibular LN | 2.33 | 2.33 | 2.88 | 2.99 | - | - | - | - | - | - |
| Hilar LN | - | - | - | 2.63 | - | - | - | - | - | - |
| Popliteal LN | 3.36 | - | - | 3.17 | - | - | - | - | - | - |
| Coronary band | - | - | - | - | - | - | - | - | - | - |
| **Probang samples** (day post challenge) | | | | | | | | | | |
| 14 | 3.27 | 3.87 | 6.21 | - | 4.17 | - | - | - | - | 4.88 |
| 17 | 6.00 | 3.84 | 5.88 | - | 4.65 | 4.39 | - | - | 4.19 | 5.32 |
| 21 | 5.74 | 5.41 | 4.63 | - | 4.71 | - | - | - | - | 5.68 |
| 24 | 5.37 | 5.03 | 4.17 | - | 4.58 | 4.44 | - | - | - | 4.63 |
| 28 | 5.80 | 4.21 | 3.77 | - | 4.29 | 5.05 | - | - | - | - |
| 31 | 4.79 | 5.80 | 3.66 | - | + | - | - | - | - | 4.24 |
| 35 | + | 4.17 | 4.33 | - | - | - | - | - | - | - |

**Fig 4. FMDV detection in ovine tissues and oropharyngeal fluid samples during persistent infection.** FMDV detection in tissue samples obtained at 35 days post challenge. The vaccinated cohort included 2 animals from each of 3 study groups subjected to different vaccination regiments; single dose vaccine at 14 days prior to virus challenge, double dose vaccine at 14 days prior to challenge, and double dose vaccine at 7 days before challenge. Highest clinical score represents the maximum cumulative lesion score observed from 0–10 days post virus challenge. Numbers in the table represent $log_{10}$ genome copy numbers (GCN)/mg of FMDV RNA in tissue or OPF samples. Color gradient indicates increasing FMDV RNA quantities in samples that were positive by both RT-qPCR and virus isolation. Numbers in uncolored cells indicate quantities of FMDV RNA that were detected in samples that were negative by virus isolation. (+) indicates that virus isolation was positive but FMDV RNA content was below the limit of detection ($<2.0$ $log_{10}$ GCN/mg), (-) indicates double-negative samples.

vaccinated animals, in which FMDV RNA-positive tissues included samples of oropharyngeal and nasopharyngeal mucosa, as well as lymph nodes draining known lesion sites (Fig 4).

**Immunomicroscopic detection of FMDV in persistently infected sheep.** FMDV structural and non-structural antigens were microscopically detected in a subset of palatine tonsil samples from the non-vaccinated group (Fig 5). The viral antigen consistently localized to cytokeratin-expressing epithelial cells within tonsillar crypts. FMDV VP1 was found as clusters of infected cells within the superficial epithelial layers in the basal area of the crypts. FMDV 3D was detected in similar regions, but as scattered individual virus-positive cells (not shown). There was no detection of FMDV antigen in any other tissues harvested during late infection.

# Discussion

Foot-and-mouth disease in sheep is often described as mild, with inconsistent clinical signs of disease [2, 4]. Nonetheless, the epidemiological relevance of sheep in relation to FMDV incursions into previously FMD-free countries was emphasized by the significant dissemination that occurred due to the movement of infected sheep during early stages of the 2001 FMD epizootic in the UK [49]. By contrast, the role of sheep in FMD epidemiology in endemic regions is largely unknown, as sheep are often not included in FMDV surveillance or vaccination programs.

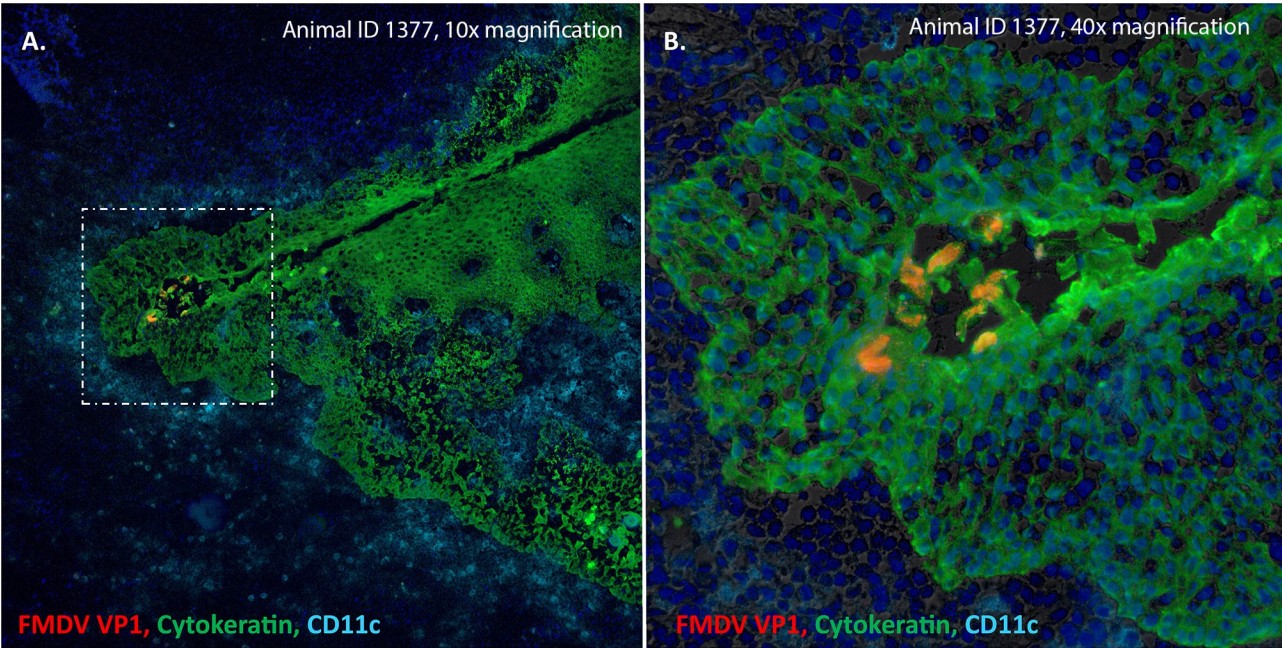

**Fig 5. Persistent FMDV infection in ovine palatine tonsil epithelium.** FMDV VP1 (red) localizes to cytokeratin+ epithelial cells (green) in superficial epithelial layers of the proximal (deep) aspect of a palatine tonsil crypt at 35 days post virus challenge (animal ID 1377). Abundant quantities of CD11c + presumptive antigen-presenting (cyan) cells are present in the surrounding tissue, but do not colocalize with viral antigen. A = 10x magnification. B = 40x magnification with differential interference contrast.

There have been several previous experimental works that have investigated FMDV infection dynamics, transmission, and vaccine efficacy in sheep [6, 33, 35, 50–53]. However, the progression and distribution of FMDV in ovine tissues during early and late phases of infection have received less attention [8, 11]. This current study was performed as an opportunistic investigation leveraging off two independently performed experiments that were designed to investigate simulated-natural inoculation systems, and heterologous vaccine protection in sheep, respectively. Subsets of sheep were selected for detailed post-mortem tissue harvest at pre-determined time points during early or persistent phases of infection. Harvested tissues were analyzed using a trimodal approach including detection of infectious virus, viral RNA, and viral antigen, and findings were interpreted in relation to similar investigations performed in cattle [26, 32] and pigs [31]. Due to the opportunistic nature of this study approach, the sample set was limited and therefore definitive or statistically proven findings were not made.

The current investigations of the early stages of FMDV infection (24–48 hours post infection) included sheep that had been infected through either intra-nasopharyngeal (INP) or aerosol inoculation. The study design also included a cohort of sheep that had been challenged by contact-exposure; however, none of the four contact-exposed sheep selected for post-mortem sampling were infected at the time of euthanasia. This is unfortunate, especially as the remaining four sheep of the same contact-exposed cohort, which were kept for clinical monitoring through 10 dpc, were all viremic at 2–3 dpc, and had clinical lesions at 3–4 dpc [5]. The sheep within the INP- and aerosol inoculated cohorts were all viremic at the time of euthanasia and tissue harvest. This confounds definitive determination of the primary infection sites as the infection was already somewhat disseminated at the time of tissue harvest. However, the majority of sheep included in the investigation still represented early stages of infection as serum FMDV RNA quantities were low, and six out of eight sheep did not yet have any detectable vesicular lesions.

INP inoculation lead to early infection of laryngopharyngeal- and oropharyngeal tonsils, with high prevalence of VI-positivity and high quantities of FMDV RNA detected in paraepiglottic- and palatine tonsils. This finding was confirmed by detection of FMDV structural and non-structural antigen within tonsil epithelium at 24 and 48 hpc. The microvesiculation of the paraepiglottic epithelium was morphologically similar to typical vesicles of lesion sites in cattle and pigs, including epithelial acantholysis and central cavitation [46]. Additionally, high FMDV RNA loads with concurrent antigen detection was found in the nasopharyngeal tonsil of one INP-inoculated individual.

In contrast, following aerosol inoculation, the highest viral RNA loads at 24 hpc were found in the lungs, with additional detection of antigen and substantial viral RNA in the nasopharyngeal mucosa. The progression of FMDV infection in aerosol-inoculated sheep is thus similar to that described for cattle following aerosol challenge [26, 28, 54]. Specifically, earlier studies of FMDV pathogenesis in cattle based on aerosol inoculation demonstrated that initial infection of the nasopharyngeal mucosa was directly followed by substantial virus amplification in the lungs [26, 28]. It was thus hypothesized that the lungs played a role in the establishment of viremia, which would seed subsequent viral replication at distant lesion sites. However, subsequent investigations utilizing INP inoculation [27] and contact challenge [32] showed that viremia and virus dissemination to distant sites occurred without substantial involvement of the lower respiratory tract. It was thereby concluded that although the bovine lungs are highly permissive to FMDV infection, viral replication in the lungs does not seem to be critical for establishment of viremia during natural exposure conditions. Based on the combined output of these studies, it is hypothesized that similar to the findings in cattle, ovine lungs can support high levels of FMDV replication. However, the extent of involvement of the lower respiratory tract in the early pathogenesis of FMDV is highly dependent on the route of virus exposure in both species. Additionally, the prominent involvement of tonsil epithelium as sites of FMDV replication during early stages of infection in INP-inoculated sheep in this current investigation is more similar to the early FMDV pathogenesis events described in pigs [31].

The paraepiglottic tonsils are paired, macroscopically visible aggregates of lymphoid tissue located within the mucosal fold at the base of the epiglottis [55]. This tonsil is present in pigs and sheep but absent in cattle [56]. In pigs, primary FMDV infection has been consistently localized to epithelial crypts of the paraepiglottic tonsil, with sustained viral amplification occurring in similar crypts of the tonsil of the soft palate (which based upon location in the oropharynx corresponds mostly to the ovine- and bovine palatine tonsils) [31, 57]. In contrast, the nasopharyngeal tonsil, which is present in cattle, sheep and pigs, is located in the upper respiratory tract, at the junction of the nasal cavity and the nasopharynx. Primary and persistent FMDV infection in cattle has consistently been localized to lymphoid-associated epithelium of the nasopharyngeal mucosa, although the nasopharyngeal tonsil is typically not affected [26, 27, 32]. The ovine- and bovine palatine tonsils are bilateral lymphoid organs embedded within a connective tissue capsule within the tissue of the soft palate, and with openings to the ventral surface of the soft palate (within the oropharynx). The epithelial crypts of the bovine palatine tonsils have, similar to the current findings, been shown to support high levels of viral replication during the clinical phase of FMD [26, 27, 32, 54, 58].

Our data confirm findings from similar studies performed in cattle which have demonstrated that the inoculation method used to infect animals has a substantial impact on the anatomic distribution of virus during early stages of infection. Although it is clear that the ovine lungs are highly susceptible to FMDV infection, similar to cattle lungs, the extent to which virus amplification occurs in the lungs during natural exposure is still unclear. Studies in cattle have confirmed that the progression of infection following INP-inoculation closely resembles the progression following natural contact exposure, in both vaccinated and naïve hosts [32].

The rapid disease progression in the sheep in the current study, combined with a lack of infection in the cohort of contact-exposed sheep selected for tissue harvest, precluded achieving similar conclusions. However, INP-inoculation of sheep has proven to be a consistent and reliable method for FMDV challenge that is applicable for both pathogenesis studies and vaccine testing in this species [5, 6].

In contrast to cattle, the only ovine tissues in which persistent FMDV was detected in carrier sheep were the paraepiglottic and palatine tonsils. The anatomic distribution of FMDV RNA was more extensive than detection of infectious virus, specifically in the non-vaccinated sheep that had undergone more severe clinical disease. This finding is similar to studies performed in both cattle and pigs, in which FMDV genomic RNA was detected with high prevalence in lymphoid tissues of convalescent animals, regardless of whether animals were persistently infected [23, 42]. This is interpreted as viral degradation products rather than ongoing persistent infection. In cattle, replicating persistent FMDV has consistently been localized to superficial layers of lymphoid-associated epithelium of the nasopharyngeal mucosa, without concurrent detection in tonsils [22, 23, 32]. Although the current findings demonstrate FMDV persistence in a morphologically similar type of lymphoid-associated epithelium of sheep, the localization to the palatine tonsil crypts is clearly distinct from cattle. However, a similar predilection for FMDV persistence in tonsil epithelium has been described in African cape buffalo [59]. In that investigation, it was also concluded that detection of FMDV in persistently infected buffalo was improved by the use of nylon brushes to access the palatine tonsil sinuses, compared to harvest of OPF by conventional probang sampling [59].

The current findings suggest that vaccination with a double dose ($>6$ $PD_{50}$ in 2 ml) of a heterologous (O1 Manisa) vaccine at 14 days prior to virus challenge conferred protection against persistent infection. This outcome was consistent across all 7 sheep in this treatment group, as demonstrated by lack of detection of FMDV in OPF at any time between 14 and 35 dpc [36]. Interestingly, this protection against FMDV persistence did not correlate with clinical protection. Specifically, both sheep from the double dose/14 day vaccination group that were selected for post-mortem sampling had vesicular lesions and fever during early stages of infection, and were thus, not protected against virus challenge. By contrast, the two sheep selected from the single dose 14 day vaccination group were clinically protected by vaccination, but were subclinically and persistently infected. The ability of high-dose vaccination to protect against FMDV persistence has been suggested by previous studies in both cattle and sheep [60–63]. However, this has most often been suggested as a direct consequence of complete protection against primary infection, which was not the case in this current investigation. Rather, the findings of this current study suggest that the high dose vaccination at 14 days prior to challenge stimulated a cell-mediated immune response [64] that was sufficient in clearing virus from the sites of primary and (potential) persistent infection.

## Conclusions

This current study has provided novel insights into the temporo-anatomic progression of FMDV infection in sheep. Noteworthy findings include the disparity of tissues identified as sites of early and persistent FMDV infection in sheep compared to cattle and pigs. Primary FMDV infection in sheep shared common elements with cattle by involvement of the nasopharyngeal mucosa, but was more similar to pigs by consistent early virus replication in the tonsils in the oropharynx and laryngopharynx. During persistent infection, FMDV was localized to epithelial crypts of the palatine tonsils, which differs from cattle, but is similar to what has been described in African buffalo. Overall, these findings emphasize critical differences in

FMDV pathogenesis across natural host species which should be considered in vaccine efficacy and field surveillance studies.

## Acknowledgments

The authors thank Elizabeth Bishop, Ethan Hartwig, George Smoliga and Steven Pauszek for processing of samples. The original FMDV isolate used was kindly provided by Dr. Kwang-Nyeong Lee, Animal and Plant Quarantine Agency, Ministry of Agriculture, Food and Rural Affairs Republic of Korea.

## Author Contributions

**Conceptualization:** Wilna Vosloo, Luis L. Rodriguez, Jonathan Arzt.

**Formal analysis:** Carolina Stenfeldt, Jonathan Arzt.

**Funding acquisition:** Wilna Vosloo, Luis L. Rodriguez, Jonathan Arzt.

**Investigation:** Carolina Stenfeldt, Juan M. Pacheco, Nagendrakumar B. Singanallur, Jonathan Arzt.

**Methodology:** Carolina Stenfeldt, Juan M. Pacheco, Nagendrakumar B. Singanallur, Jonathan Arzt.

**Project administration:** Wilna Vosloo, Luis L. Rodriguez, Jonathan Arzt.

**Supervision:** Wilna Vosloo, Luis L. Rodriguez, Jonathan Arzt.

**Validation:** Carolina Stenfeldt, Juan M. Pacheco, Wilna Vosloo, Luis L. Rodriguez, Jonathan Arzt.

**Visualization:** Carolina Stenfeldt.

**Writing – original draft:** Carolina Stenfeldt.

**Writing – review & editing:** Juan M. Pacheco, Nagendrakumar B. Singanallur, Wilna Vosloo, Luis L. Rodriguez, Jonathan Arzt.

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
