## [Decision Letter · Decision Letter 0]

19 Nov 2019

PONE-D-19-28136

Virulence beneath the fleece; a tale of foot-and-mouth disease virus pathogenesis in sheep

PLOS ONE

Dear Dr. Stenfeldt,

Thank you for submitting your manuscript to PLOS ONE. After careful consideration, we feel that it has merit but does not fully meet PLOS ONE’s publication criteria as it currently stands. Therefore, we invite you to submit a revised version of the manuscript that addresses the points raised during the review process.

 Please pay close attention to the concerns raised by reviewer # 2. You are requested to address the issues raised by both the reviewers with special emphaisis with issues raised by reviewer # 2.

We would appreciate receiving your revised manuscript by Jan 03 2020 11:59PM. To enhance the reproducibility of your results, we recommend that if applicable you deposit your laboratory protocols in protocols.io, where a protocol can be assigned its own identifier (DOI) such that it can be cited independently in the future. For instructions see: http://journals.plos.org/plosone/s/submission-guidelines#loc-laboratory-protocols

We look forward to receiving your revised manuscript.

Kind regards,

Aftab A. Ansari, PhD

Academic Editor

PLOS ONE

Journal Requirements:

Reviewers' comments:

Reviewer's Responses to Questions

**Comments to the Author**

1. Is the manuscript technically sound, and do the data support the conclusions?

Reviewer #1: Yes

Reviewer #2: Yes

2. Has the statistical analysis been performed appropriately and rigorously? 

Reviewer #1: No

Reviewer #2: N/A

3. Have the authors made all data underlying the findings in their manuscript fully available?

Reviewer #1: Yes

Reviewer #2: Yes

4. Is the manuscript presented in an intelligible fashion and written in standard English?

Reviewer #1: Yes

Reviewer #2: Yes

5. Review Comments to the Author

Reviewer #1: Carolina Stenfeldt et al. explored the tissue- and cellular tropism of early and late stages of FMDV infection in sheep following different routes of simulated natural virus challenge. It was determined that the paraepiglottic- and palatine tonsils, as well as the nasopharyngeal mucosa were susceptible to primary FMDV infection in sheep. In addition, the authors found that the persistent FMDV infection is located to the paraepiglottic- and palatine tonsils in sheep, which is different with that in cattle. A strict epithelial tropism was identified during both early and late phases of infection. A large amount of meaningful studies on FMDV pathogenesis in pigs and cattle had been conducted by Professor Jonathan Arzt’s lab previously. This study provided more detailed information and insight in FMDV pathogenesis in sheep. Some clarifications will improve the manuscript.

1) The authors need to describe in detail in the Materials and Methods section about the statistical analysis method.

2) I think Figure 1 should be converted to a table. So does Figure 4.

3) The flaw of this study is that there were only two animals in each group. The significance of the results was weak. How did the authors make conclusion when there were different results in two animals. Such as, the comparably high viral RNA quantities could be detected in the Submandibular LN in animal 1301, but not in that of 1302.

4) I do believe that some conclusions could only be made when the viral distribution was consistent in both of the two infected animals.

5) The virus replication was extremely high in animal 1311, however no fever was observed. Was there a correlationship between the fever occurrence and viral replication level at 48 h post challenge?

6) Magnification and animal number should be marked in Figure 2, 3 and 5 but not just in the legends.

7) Although the authors have mentioned the lack of detection of FMDV RNA in the dose/14 day vaccination group with clinical manifestation of FMD, more discussion should be added in the text.

8) Define acronyms the first time one appears in the text.

9) Could the authors summarize the disparity of tissues of infection sites among different animals in a table? It will make present data more meaningful and easily understood.

Reviewer #2: The manuscript by Stenfeldt et al. presents relevant and detailed information on the FMDV pathogenesis in sheep, working both at early and later times post-infection with different experimental infection protocols, in naive and vaccinated animals (immunized with a heterologous vaccine strain). Interestingly, these results demonstrate that while the early progression of the infection shares similarities with cattle, the persistence infection occurs in different tissues for ovine species, involving palatine tonsils (as described in buffalos). Similarly interesting is the finding that early infection pattern was also similar to pigs due to the early virus replication in the tonsils in the oropharynx and laryngopharynx. It is also important to note that, as previously described for cattle and swine, microscopic FMDV infection seems tightly restricted to epithelial cells in mucosal tissues, at both early and late infection times.

Comments and questions:

- It is interesting to note that animal #1301, which showed a generalization of the symptoms (fig, 1), was also the one to present the highest levels of FMDV detection in serum and the lower respiratory tissues (lung). Is it possible to hypothesize that the lungs may act as a portal to amplify viral titers, which in turn may allow the infective virus to reach distal tissues (such the interdigital cleft and coronary bands) through the peripheral circulation? Since this has been already proposed for cattle, it may be worth to mention such a hypothesis also in the discussion.

- One of the INP-infected animals showed a correlation between the existence of macroscopic lesions on the dental pad and in the interdigital cleft, with the detection of viral RNA in both tissues (4.50 and 7.40, respectively). However, none of the aerosol infected animals exhibited macroscopic lesions in any of these lesion predilection sites in spite that both individuals showed high levels of viral RNA detection (6.59/8.56 and 3.98/7.85, respectively). Is it possible to relate such differential progression of the symptoms to the route infection?

- What are the potential physical or anatomical explanations for the differential levels of infection in the dorsal nasopharyngeal mucosa in those animals infected by INP- or aerosol-inoculation?

- In figure 4, animals receiving a double dose of FMD vaccines showed (mild) clinical scores while those receiving a single vaccine dose did not (both groups challenged at 14 dpv). However, as stated in the discussion, all individuals receiving double vaccine dose were negative to the detection and isolation of FMDV RNA up to 35 dpc. Do the authors have any hypothesis on how a prophylactic treatment could prevent persistent infection but not clinical symptoms?

6. PLOS authors have the option to publish the peer review history of their article (what does this mean?). If published, this will include your full peer review and any attached files.

Reviewer #1: No

Reviewer #2: Yes: Mariano Perez-Filgueira

---

## [Author Response · Author response to Decision Letter 0]

3 Dec 2019

Reviewer #1: Carolina Stenfeldt et al. explored the tissue- and cellular tropism of early and late stages of FMDV infection in sheep following different routes of simulated natural virus challenge. It was determined that the paraepiglottic- and palatine tonsils, as well as the nasopharyngeal mucosa were susceptible to primary FMDV infection in sheep. In addition, the authors found that the persistent FMDV infection is located to the paraepiglottic- and palatine tonsils in sheep, which is different with that in cattle. A strict epithelial tropism was identified during both early and late phases of infection. A large amount of meaningful studies on FMDV pathogenesis in pigs and cattle had been conducted by Professor Jonathan Arzt’s lab previously. This study provided more detailed information and insight in FMDV pathogenesis in sheep. Some clarifications will improve the manuscript.

1) The authors need to describe in detail in the Materials and Methods section about the statistical analysis method.

Response: Statistical analyses were not performed in this study. As noted by the reviewer (in comment 3) the group sizes available for comparison were small, and the output of the investigation is therefore purely of a descriptive nature, and not suitable for statistical analyses. It is our hope to have more sheep-derived data in coming years in order to characterize ovine FMD with statistical significance. Despite this clear limitation, we are confident that the novel descriptive information in this paper which will be a stepwise contribution towards further elucidation of FMD in sheep. 

Our research group has an established record of previous publications based on similarly descriptive (primary) studies describing FMDV pathogenesis in different host species (DOI: 10.1177/0300985810372509, 10.1371/journal.pone.0106859, 10.1371/journal.pone.0143666). We have also demonstrated that the output of such (primary) studies may subsequently be utilized for quantitative analyses in which statistical output and conclusions can be made (DOI: 10.3389/fvets.2019.00263, 10.3389/fvets.2018.00167, 10.1038/s41598-019-39029-0).

We have added a sentence to the methods section (lines 126-128) to clarify that statistical analyses were not performed. Additionally, we have now added tabulated data as a supplemental file, so that other investigators may be able to incorporate these data into quantitative analyses. 

2) I think Figure 1 should be converted to a table. So does Figure 4.

Response: As suggested by the reviewer, we have now added the same data from Fig 1 & 4 as a supplemental table to provide access for readers who prefer that form. However, in the manuscript we have chosen to include the presented material as figures rather than tables to allow the use of a color gradient to facilitate readers’ rapid interpretation of the results within the manuscript. Specifically, the color gradient allows for an easier overview of anatomic regions with specifically high- or low virus detection. 

3) The flaw of this study is that there were only two animals in each group. The significance of the results was weak. How did the authors make conclusion when there were different results in two animals. Such as, the comparably high viral RNA quantities could be detected in the Submandibular LN in animal 1301, but not in that of 1302.

Response: As mentioned above (response #1), we agree regarding the limitations associated with small group sizes in these studies. On this basis, we are not suggesting statistical significance of any findings anywhere in the manuscript. Rather, we have been careful to be clear that the study is purely descriptive. In such instances as pointed out regarding the different levels of virus detection in the submandibular lymph node of animals 1301 and 1302, we believe that the real conclusion is that it is common that in natural host species biological variation (including variation in the timing of disease progression in individual animals) will mean that it is not always possible to make clear interpretations. Based on our previous experiences with generating and reporting this type of data, we have chosen to emphasize higher level patterns, such as the apparent difference in virus localization to the lower respiratory tract during early infection, in animals infected via aerosol- versus intra-nasopharyngeal inoculation.

4) I do believe that some conclusions could only be made when the viral distribution was consistent in both of the two infected animals.

Response: It is not entirely clear if the reviewer is responding to a specific line item or rather making a general comment. As mentioned above, we believe that it is an unavoidable and useful output to demonstrate that even in controlled experiments, individual-animal variability is part of natural pathogenesis and disease progression. As has now been clarified in the methods (lines 126-128) and discussion (lines 383-385), we are not claiming statistical significance of any of the presented findings.

5) The virus replication was extremely high in animal 1311, however no fever was observed. Was there a correlationship between the fever occurrence and viral replication level at 48 h post challenge?

Response: As correctly pointed out by the reviewer, there was no clear correlation between the detection of fever, and levels of viral replication in tissues. We have found this to often be the case through multiple studies of FMDV pathogenesis in ruminants, mostly cattle. 

6) Magnification and animal number should be marked in Figure 2, 3 and 5 but not just in the legends.

Response: We have published many papers in PLOS including microscopy images, and we are confident that such information is specified to be included in the legends. However, if the editors indicate that such information should be added to the image plates, then we will comply. 

7) Although the authors have mentioned the lack of detection of FMDV RNA in the dose/14 day vaccination group with clinical manifestation of FMD, more discussion should be added in the text.

Response: As mentioned within the manuscript, the animals used for study of the FMDV carrier state in sheep were derived from a vaccine challenge study that has been previously published (doi: 10.1016/j.antiviral.2017.07.020) and only a subset of the experimental animals from those studies were described in further detail herein. For these reasons, we feel it would not be scientifically sound or appropriate to extrapolate further or over-interpret these findings based on the limitations of the data set.

However, we have inserted one additional sentence at the end of the discussion (489-491) speculating on potential mechanisms of virus clearance in these animals. 

8) Define acronyms the first time one appears in the text.

Response: This has been reviewed and accomplished as per reviewer’s suggestion.

9) Could the authors summarize the disparity of tissues of infection sites among different animals in a table? It will make present data more meaningful and easily understood.

Response: As mentioned above, due to limited sample sizes and individual animal variation, we believe it would not be appropriate to make such conclusions based on the available data. The color coding in Figure 1 is intended to emphasize higher level patterns, such as the disparity between the groups in the extent of virus replication in the lower respiratory tract at 24 hpi. That specific finding represents the most substantial difference in virus distribution between groups, as has been emphasized in results (250-251) and discussion (408-426, 442-453)

Reviewer #2: The manuscript by Stenfeldt et al. presents relevant and detailed information on the FMDV pathogenesis in sheep, working both at early and later times post-infection with different experimental infection protocols, in naive and vaccinated animals (immunized with a heterologous vaccine strain). Interestingly, these results demonstrate that while the early progression of the infection shares similarities with cattle, the persistence infection occurs in different tissues for ovine species, involving palatine tonsils (as described in buffalos). Similarly interesting is the finding that early infection pattern was also similar to pigs due to the early virus replication in the tonsils in the oropharynx and laryngopharynx. It is also important to note that, as previously described for cattle and swine, microscopic FMDV infection seems tightly restricted to epithelial cells in mucosal tissues, at both early and late infection times.

Comments and questions:

- It is interesting to note that animal #1301, which showed a generalization of the symptoms (fig, 1), was also the one to present the highest levels of FMDV detection in serum and the lower respiratory tissues (lung). Is it possible to hypothesize that the lungs may act as a portal to amplify viral titers, which in turn may allow the infective virus to reach distal tissues (such the interdigital cleft and coronary bands) through the peripheral circulation? Since this has been already proposed for cattle, it may be worth to mention such a hypothesis also in the discussion.

Response: This is a valid point. We have inserted a new passage in the discussion (line 410-425) to expand upon this concept. In brief, the FMDV tissue distribution in aerosol inoculated sheep in the current investigation was highly similar to previous publications describing FMDV pathogenesis in cattle infected by aerosol exposure. However, in our subsequent studies of FMDV pathogenesis in cattle using intra-nasopharyngeal (INP) deposition or contact exposure, we found that viral amplification in the lungs during early infection did not occur. Specifically, based on findings from our contact exposure studies in cattle, we have concluded that although the lungs are clearly permissive to high levels of FMDV replication, this does not seem to be necessary for establishment of viremia or dissemination of virus to distant replication sites.

- One of the INP-infected animals showed a correlation between the existence of macroscopic lesions on the dental pad and in the interdigital cleft, with the detection of viral RNA in both tissues (4.50 and 7.40, respectively). However, none of the aerosol infected animals exhibited macroscopic lesions in any of these lesion predilection sites in spite that both individuals showed high levels of viral RNA detection (6.59/8.56 and 3.98/7.85, respectively). Is it possible to relate such differential progression of the symptoms to the route infection?

Response: We cannot rule out this possibility, however we do not believe that these differences were caused by the different routes of virus inoculation, but rather a consequence of different timing of progression of infection. Thus, we believe that the high levels of virus replication found in lesion predilection sites are indicative of lesions developing at these sites, even though macroscopic lesions were not yet detectable in some of the animals. High levels of viremia at those timepoints also surely contributed to the widespread tissue-level detection in those animals. We have inserted a sentence in the results section (260-261) to clarify these points.

- What are the potential physical or anatomical explanations for the differential levels of infection in the dorsal nasopharyngeal mucosa in those animals infected by INP- or aerosol-inoculation?

Response: It is difficult to make any clear conclusions on this based on the limited number of animals included in the study. That said, there is a clear distinction in the physical properties of the initial inoculation schemes; INP�just nasopharynx; aerosol�entire respiratory tract including nasopharynx (upper respiratory tract). This is why it is not surprising that the aerosol-inoculated sheep have distribution of FMDV in nasopharynx AND lungs from early timepoints, whereas INP-inoculated sheep have more limited distribution at early time points. 

 - In figure 4, animals receiving a double dose of FMD vaccines showed (mild) clinical scores while those receiving a single vaccine dose did not (both groups challenged at 14 dpv). However, as stated in the discussion, all individuals receiving double vaccine dose were negative to the detection and isolation of FMDV RNA up to 35 dpc. Do the authors have any hypothesis on how a prophylactic treatment could prevent persistent infection but not clinical symptoms?

 Response: We agree that this is an interesting finding; and it is clear that the current data set is only sufficient for “hypothesis-generating”. Based on the full data set from the vaccine challenge study (including the animals that were not used for postmortem tissue harvest), it seems that all vaccinated animals were indeed infected (regardless of clinical protection) as is expected with FMDV vaccines. Additionally, all of the vaccinated groups included animals that were clinically protected, and those that were not. However, all animals in the double dose/14 day vaccine dose were efficient in complete clearance of virus as there were no carriers in that group (the full outcome of the vaccine challenge study is published in doi: 10.1016/j.antiviral.2017.07.020). 

Based on previous work done in cattle (DOI:10.1038/s41598-017-18112-4), we hypothesize that efficient clearance of FMDV from infected animals is associated with stimulation of cell-mediated immunity. This contrasts conventional wisdom that the clearance of FMD in the clinical phase is mostly dependent on actuation of the TH2-asociated (antibody-mediated) response. 

Without more information from the current study, we cannot draw any substantive conclusions of mechanisms involved in the current case. However, we have inserted one additional sentence at the end of the discussion (489-491) speculating on potential mechanisms.

---

## [Editor Report · Decision Letter 1]

6 Dec 2019

PONE-D-19-28136R1

Virulence beneath the fleece; a tale of foot-and-mouth disease virus pathogenesis in sheep

PLOS ONE

Dear Dr. Stenfeldt,

Thank you for submitting your manuscript to PLOS ONE. After careful consideration, we feel that it has merit but does not fully meet PLOS ONE’s publication criteria as it currently stands. Therefore, we invite you to submit a revised version of the manuscript that addresses the points raised during the review process.

   As stated in the revised critique, the authors have done an excellent job in addressing all the concerns rasied by the reviewers except two minor issues remain. First of all, the same data that appear in Fig. 1 and 4 do not need to be re-expressed in a Table form even if it is part of the supplementary data section as they are redundant and therefore the 2 supplementary tables are recommended for deletion. In addition, the authors are kindly requested to add the names of the animals and the magnification in Fig. 2, 4 and 5 as it greatly assists readers in following the narrative.

We would appreciate receiving your revised manuscript by Jan 20 2020 11:59PM. To enhance the reproducibility of your results, we recommend that if applicable you deposit your laboratory protocols in protocols.io, where a protocol can be assigned its own identifier (DOI) such that it can be cited independently in the future. For instructions see: http://journals.plos.org/plosone/s/submission-guidelines#loc-laboratory-protocols

We look forward to receiving your revised manuscript.

Kind regards,

Aftab A. Ansari, PhD

Academic Editor

PLOS ONE

Additional Editor Comments (if provided):

I believe the authors have done an excellent job in addressing the concerns of the reviewers. However, two minor issues remain. First of all, I think that the addition of the same data in Table format for Figs. 1 and 4 is redundant. The authors should delete the Supplementary tables. Secondly, I do think that the addition of the animal numbers in the figures makes it easier to follow the data and I strongly urge the authors to insert the animal numbers and magnification in Fig. 2, 3 and 5.

---

## [Author Response · Author response to Decision Letter 1]

8 Dec 2019

I believe the authors have done an excellent job in addressing the concerns of the reviewers. However, two minor issues remain. First of all, I think that the addition of the same data in Table format for Figs. 1 and 4 is redundant. The authors should delete the Supplementary tables. Secondly, I do think that the addition of the animal numbers in the figures makes it easier to follow the data and I strongly urge the authors to insert the animal numbers and magnification in Fig. 2, 3 and 5.

Response: The supporting information files have been discarded. Animal IDs and magnification have been added to the image panels as suggested.

---

## [Editor Report · Decision Letter 2]

12 Dec 2019

Virulence beneath the fleece; a tale of foot-and-mouth disease virus pathogenesis in sheep

PONE-D-19-28136R2

Dear Dr. Stenfeldt,

We are pleased to inform you that your manuscript has been judged scientifically suitable for publication and will be formally accepted for publication once it complies with all outstanding technical requirements.

With kind regards,

Aftab A. Ansari, PhD

Academic Editor

PLOS ONE
---

## [Editor Report · Acceptance letter]

16 Dec 2019

PONE-D-19-28136R2 

Virulence beneath the fleece; a tale of foot-and-mouth disease virus pathogenesis in sheep 

Dear Dr. Stenfeldt:

I am pleased to inform you that your manuscript has been deemed suitable for publication in PLOS ONE. Congratulations! Your manuscript is now with our production department. 

With kind regards,

on behalf of

Dr. Aftab A. Ansari 

Academic Editor

PLOS ONE